# ATAC-Seq Identifies Chromatin Landscapes Linked to the Regulation of Oxidative Stress in the Human Fungal Pathogen *Candida albicans*

**DOI:** 10.3390/jof6030182

**Published:** 2020-09-21

**Authors:** Sabrina Jenull, Michael Tscherner, Theresia Mair, Karl Kuchler

**Affiliations:** Medical University of Vienna, Center for Medical Biochemistry, Max Perutz Labs Vienna, Campus Vienna Biocenter, Dr. Bohr-Gasse 9/2, A-1030 Vienna, Austria; sabrina.jenull@meduniwien.ac.at (S.J.); michael.tscherner@univie.ac.at (M.T.); theresia.mair@meduniwien.ac.at (T.M.)

**Keywords:** candida, chromatin, oxidative stress, ATAC-seq

## Abstract

Human fungal pathogens often encounter fungicidal stress upon host invasion, but they can swiftly adapt by transcriptional reprogramming that enables pathogen survival. Fungal immune evasion is tightly connected to chromatin regulation. Hence, fungal chromatin modifiers pose alternative treatment options to combat fungal infections. Here, we present an assay for transposase-accessible chromatin using sequencing (ATAC-seq) protocol adapted for the opportunistic pathogen *Candida albicans* to gain further insight into the interplay of chromatin accessibility and gene expression mounted during fungal adaptation to oxidative stress. The ATAC-seq workflow not only facilitates the robust detection of genomic regions with accessible chromatin but also allows for the precise modeling of nucleosome positions in *C. albicans*. Importantly, the data reveal genes with altered chromatin accessibility in upstream regulatory regions, which correlate with transcriptional regulation during oxidative stress. Interestingly, many genes show increased chromatin accessibility without change in gene expression upon stress exposure. Such chromatin signatures could predict yet unknown regulatory factors under highly dynamic transcriptional control. Additionally, de novo motif analysis in genomic regions with increased chromatin accessibility upon H_2_O_2_ treatment shows significant enrichment for Cap1 binding sites, a major factor of oxidative stress responses in *C. albicans*. Taken together, the ATAC-seq workflow enables the identification of chromatin signatures and highlights the dynamics of regulatory mechanisms mediating environmental adaptation of *C. albicans*.

## 1. Introduction

Human fungal pathogens respond to host immune defense through numerous mechanisms, including chromatin-mediated adaptive gene expression [1,2,3]. For example, immune defense or environmental changes can trigger pathogen responses through extracellular sensing, intracellular signal transduction, and transcriptional reprogramming [4,5]. Transcriptional changes require a tight interplay of chromatin states and transcription factors [6,7], as the swift adaptation to environmental changes is often paramount for a successful lifestyle or immune evasion. For instance, pathogens encounter a number of extreme stress conditions during the course of an infection. These include limitations in nutrient availability and the cytotoxic attack by the host immune system [8,9]. The opportunistic fungal pathogen *Candida albicans* is an extraordinary example of how a pathogen can occupy multiple host niches to persist and survive. *C. albicans* lives as a harmless commensal in the majority of humans, colonizing mucosal surfaces and epithelial barriers, especially the intestinal tract. However, severe immunodeficiency or a dysregulation of the host microbiota turns *C. albicans* into an invasive pathogen that can infect virtually any tissue or organ in the human body [10,11]. In the last decade, numerous efforts have been made to better understand fungal pathogenicity mechanisms, including the transcriptional regulation of environmental adaptation and virulence factors, such as the switch between different cellular morphologies [12,13,14]. Given the pivotal interplay of chromatin modifications and transcription control, it is not surprising that several *C. albicans* chromatin-modifying factors play important roles in fungal virulence. For instance, the functions of the histone acetyl transferases (HATs) Gcn5, Hat1, and Rtt109 are crucial for morphogenesis and virulence [15,16,17,18,19]. Likewise, the histone deacetylase (HDAC) complex Set3C controls transcriptional kinetics during the morphological transition from yeast growth to filamentous hyphal growth. Remarkably, genetic ablation of *SET3* abrogates fungal virulence [20]. Hence, attacking chromatin modifiers provides a new option for antifungal drug development. This requires immediate attention in clinical settings, given the rapid emergence of antifungal drug resistance in *Candida* spp. such as *C. glabrata* or *C. auris* [17,21,22,23,24,25]. However, a better understanding of the interplay between chromatin architecture in the pathogen and transcriptional reprogramming during host interaction would further aid the discovery of new antifungals targeting chromatin function.

Several methods including chromatin immunoprecipitation (ChIP) and micrococcal nuclease (MNase) digestion of chromatin coupled with next-generation sequencing (ChIP-seq and MNase-seq, respectively) or DNase-seq have been employed to analyze chromatin accessibility and nucleosome positioning to reveal regulatory mechanisms [26,27,28,29,30]. Recently, the assay for transposase-accessible chromatin using sequencing (ATAC-seq) emerged to probe for native chromatin states, which includes accessibility as well as nucleosome positioning. ATAC-seq employs a hyperactive Tn5 transposase loaded with sequencing adapters, which are inserted into accessible chromatin sites, causing fragmentation and tagging of chromatin DNA referred to as tagmentation. This happens preferably at genomic regions with open, accessible chromatin, since transposition events into condensed chromatin are less likely [30,31]. Due to its technical simplicity and the low sample input requirements, ATAC-seq has been widely applied for chromatin profiling in various cell types [31,32,33,34], tissues [35,36], and even single cells [37,38]. Moreover, it has proved to be a useful tool for identifying sequence motifs decorated by transcriptional regulators and for predicting gene transcription [39,40,41,42].

Here, we aimed to adapt the original ATAC-seq protocol [30] for the human fungal pathogen *C. albicans.* We developed a modified protocol and bioinformatics workflow that enables the sensitive detection of changes in the global chromatin landscapes in response to environmental stress. We have chosen oxidative stress as environmental cue, because it triggers genome-wide changes in gene expression [14,16] and because it closely mimics the oxidative immune defense fungal pathogens face during host invasion [43,44]. Moreover, transcriptomics data for *C. albicans* challenged with hydrogen peroxide (H_2_O_2_) and genome-wide binding data of the oxidative stress transcriptional regulator Cap1 are available [16,45], which we combined with the present ATAC-seq data for *C. albicans*. With this approach, we demonstrate that H_2_O_2_-treatment of fungal cells increases chromatin accessibility in upstream regions of genes associated with the oxidative response, and we show that those genes tend to be transcriptionally upregulated. Moreover, signatures of accessible chromatin regions enable the prediction of putative novel regulators of stress signaling that are not yet linked to transcriptional control. In addition, genomic regions with an elevated ATAC-seq signal are enriched in binding sites for the key regulator Cap1, demonstrating the potential for de novo motif discovery of regulatory factors. In summary, we show the versatility of ATAC-seq chromatin profiling in *C. albicans*, especially when combined with complementary next-generation sequencing data such as RNA-seq. This approach uncovers dynamic and complex regulatory mechanisms during environmental adaptation of pathogens.

## 2. Materials and Methods

### 2.1. Fungal Strains, Media, and Growth Conditions

The *C. albicans* mating type-like loci (MTL) a/α clinical isolate SC5314 [46] was used for all experiments and was routinely grown in YPD (1% yeast extract, 2% peptone, 2% glucose; all BD Biosciences) at 30 °C. 

To induce oxidative stress, *C. albicans* was first grown overnight to an optical density at 600 nm (OD_600_) of approximately 1 in YPD at 30 °C and was then treated with 1.6 mM H_2_O_2_ (Sigma-Aldrich, Burlington, MA, USA) for 15 min while shaking at 30 °C. As control, cells without H_2_O_2_ treatment were cultured in parallel. Cells were harvested and directly subjected to spheroplasting and tagmentation (see below).

### 2.2. Nuclei and Genomic DNA (gDNA) Isolation, Tagmentation, ATAC-Seq Libraries, and Sequencing

Nuclei preparation for tagmentation from *C. albicans* grown in YPD +/− H_2_O_2_ was based on a previously published protocol used for *Saccharomyces cerevisiae* [31]. Briefly, 3 × 10^7^
*C. albicans* cells were washed 1× in sorbitol buffer (1.4 M sorbitol, 40 mM HEPES-KOH pH 7.5, 0.5 mM MgCl_2_, 10 mM DTT; all Sigma-Aldrich). After cell harvesting, the pellet was then resuspended in 330 µL sorbitol buffer and 192 µL of this suspension was mixed with 8 µL of 50 mg/mL 100T Zymolase (2 mg/mL final concentration; Sigma-Aldrich) and incubated for 5 min at 30 °C. Spheroplasting efficiency was monitored by diluting a 10 µL aliquot of the spheroplasting reaction into 1 mL distilled water and OD_600_ measurement. After 5 min of spheroplasting, the OD_600_ dropped by >90% with respect to the initial OD_600_. To sustain oxidative stress for H_2_O_2_-treated cells, 1.6 mM H_2_O_2_ was added to the spheroplasting reaction. Hence, the total time of H_2_O_2_ exposure was 20 min. Spheroplasts were then harvested for 5 min at 2000× *g* at 4 °C and washed 1× in ice-cold sorbitol buffer without DTT. The pellet was resuspended in 500 µL ice-cold sorbitol buffer without DTT and spheroplasts were counted on a CASY^®^ cell counter. Five million spheroplasts were transferred into a tube on ice, harvested for 2 min at 2000× *g* at 4 °C and the resulting cell pellet was used for subsequent tagmentation.

#### 2.2.1. gDNA Isolation

To control for the sequence bias of the Tn5 transposase (TDE1) [47], gDNA was isolated via phenol–chloroform–isoamyl alcohol (PCI; Sigma-Aldrich) extraction from stationary-phase cells from *C. albicans* as described earlier [48] with slight modifications. Briefly, cells were broken up in lysis buffer [48] using a FastPrep instrument (MP Biomedicals, Santa Ana, CA, USA; settings: 2 rounds of 45 s 6 m/s with 5 min on ice in between). After PCI extraction, gDNA was precipitated with 100% ethanol, treated with 10 mg/mL RNase A (Sigma-Aldrich) and again precipitated with ammonium acetate and 100% ethanol. The final DNA pellet was resuspended in 50 µL TE (10 mM Tris-HCl pH 8, 1 mM EDTA) and subjected to the ATAC-seq library preparation workflow described below. The concentration was measured with QuantiFluor (Promega, Madison, WI, USA) according to the manufacturer’s instructions.

#### 2.2.2. Tagmentation

Five million fungal spheroplasts or 0.5 ng naked gDNA were resuspended in the tagmentation reaction mix (12.50 μL Nextera 2× TD buffer, 2.00 µL Nextera TDE1 (all Illumina), 0.50 µL 50× protease inhibitor cocktail (Roche, Basel, Switzerland), 0.25 µL 1% Digitonin (New England Biolabs, Ipswich, MA, USA, Frankfurt am Main, Germany) and 10.25 µL nuclease-free distilled H_2_O (ThermoFisher Scientific, Shanghai, China) and incubated at 37 °C for 30 min. The tagmentation reaction was immediately purified using a Qiagen MiniElute PCR purification kit and elution was done in 12 µL elution buffer provided by the kit.

#### 2.2.3. ATAC-seq Library Amplification and Size Selection

ATAC-seq library preparation was based on the protocol published by the Greenleaf Lab [49] with minor modifications. Briefly, prior to PCR amplification of the ATAC-seq libraries, a test qPCR was performed to determine the optimal number of amplification cycles in order to avoid size and GC base bias of the ATAC-seq library [49]. The qPCR contained the following components: 1 µL tagmented DNA, 0.5 µL Nextera Index primer 1 noMX and Index primer 2.1. barcode (25 µM each), 0.1 µL 100× SYBR green (Sigma-Aldrich, freshly diluted from a 10,000× stock), 5 µL 2× NEBnext High-Fidelity PCR Master Mix (New England Biolabs), and 2.9 µL nuclease-free distilled H_2_O (ThermoFisher Scientific). The qPCR was performed in a Realplex Mastercycler (Eppendorf, Hamburg, Germany) with the following cycling conditions: 72 °C 5 min, 98 °C 30 s, 25 cycles of 98 °C 10 s, 63 °C 30 s, 72 °C 1 min and a final hold step at 10 °C. The cycle number for optimal ATAC-seq library amplification was determined as in [49]. The enrichment PCR was then performed using 10 µL tagmented DNA, 2.5 µL Index primer 1 noMX and Index primer 2.x. barcode (25 µM each), 25 µL 2× NEBnext High-Fidelity PCR Master Mix (New England Biolabs) and 10 µL nuclease-free distilled H_2_O (ThermoFisher Scientific). The PCR reaction was then incubated in a PCR thermocycler using the following conditions: 72 °C 5 min, 98 °C 30 s, x cycles (depending on the test qPCR results) of 98 °C 10 s, 63 °C 30 s, 72 °C 1 min, and a final hold step at 10 °C. All ATAC-seq libraries from this study were PCR amplified using 11 cycles. For multiplexing, a different Nextera Index primer 2 barcode was used for each sample. See Appendix A for a list of Nextera PCR primers used for library amplification in this study.

The amplified ATAC-seq libraries were immediately purified and size selected with a double-sided solid-phase reversible immobilization (SPRI) approach (0.5×/1.4×) using AMPure XP beads (Beckman Coulter, Pasadena, CA, USA) according to the manufacturer’s instructions. Final DNA elution was done with elution buffer provided in the Qiagen MiniElute PCR purification kit. The final ATAC-seq libraries were quantified using the fluorescent dye QuantiFluor (Promega) according to the manufacturer’s manual. The yield for each ATAC-seq library sample was between 60 and 90 nM for replicate 1, replicate 2 and around 13 nM for replicate 3.

The quality of purified libraries was analyzed on a Bioanalyzer High Sensitivity DNA chip (Agilent, Santa Clara, CA, USA; see Figure 1B for an example) by following the manufacturer’s instructions.

#### 2.2.4. Next-Generation Sequencing

ATAC-seq libraries were prepared from three biological replicates for each condition (YPD, YPD + H_2_O_2_, and gDNA) and pooled in equimolar ratios for sequencing. Sequencing was done in 75 bp paired-end read mode on a HiSeq 3000/4000 system at the Biomedical Sequencing Facility (BSF; https://cemm.at/research/facilities/biomedical-sequencing-facility-bsf/) at the Center of Molecular Medicine (CeMM) in Vienna, Austria.

### 2.3. ATAC-Seq Data Analysis Workflow

#### 2.3.1. Pre-Processing and Read Alignment

Quality control of raw sequencing read files (.bam) was done using fastQC v0.11.8 [50]. Illumina TrueSeq adapter trimming was done via cutadapt v1.18 (https://cutadapt.readthedocs.io/en/stable/; settings: –interleaved-q 30 -O 1). Trimmed reads were then aligned to the haploid *C. albicans* SC5314 genome (assembly 22, version A22-s07-m01-r88; http://www.candidagenome.org/) using NextGenMap v0.5.5 [51] only keeping aligned reads with a minimum mapping quality of 30 (settings: -b –p -Q 30). Optical read duplicates were removed using Picard tools (Broad Institute, https://broadinstitute.github.io/picard/, settings: MarkDuplicates REMOVE_DUPLICATES=true VALIDATION_STRINGENCY=LENIENT; Broad Institute, https://broadinstitute.github.io/picard/) and mitochondrial reads were removed using the “intersect” tool from BEDTools with –v settings (https://github.com/arq5x/bedtools2). The average number of mapped reads per conditions for each chromosome and the size of each chromosome in comparison are shown in Appendix A. Chromosome sizes of *C. albicans* were retrieved from the National Center for Biotechnology Information (NCBI). The ATAC-seq fragment length distribution from properly paired reads is represented in Figure 1C.

Aligned BAM files were split according to the fragment lengths of the sequencing read pairs as done previously [30]. Read fragments below 100 bp were considered as coming from nucleosome-free regions and were used for further analysis. Normalized read coverage files (bigWig) of nucleosome-free read fragments were created using deepTools2 “bamCoverage” ([52]; settings: -e -bs 5 --normalizeUsing CPM) and visualized using the Integrative Genomics Viewer (IGV; [53]).

#### 2.3.2. Prediction of Nucleosomal Positions and Genomic Coverage of ATAC-seq Signals

Nucleosomal occupancies were predicted using NucleoATAC [31] with default parameters. NucleoATAC analysis was performed for all *C. albicans* promoter regions (transcription start site (TSS) −/+ 1000 bp). Promoter regions were extracted using the “promoters” function from the Bioconductor GenomicRanges package [54]. The NucleoATAC bedgraph output files were further converted to the bigWig format using the bedGraphToBigWig tool (https://github.com/ENCODE-DCC/kentUtils). The NucleoATAC result was further compared with the nucleosome-free ATAC-seq reads (see above) and published MNase-seq data for *C. albicans* grown in YPD [55]. MNase-seq raw datasets were downloaded via sequence read archive (SRA) and processed as the ATAC-seq raw data. The aligned BAM files were converted into a normalized read coverage (bigWig) file using the deepTools2 “bamCoverage” function ([52], settings: as above, except −bs 1). Genomic read coverage tracks were visualized with the IGV. To plot the read coverage of nucleosome-free ATAC-seq peaks, nucleosome positions from the NucleoATAC analysis, and the published MNase-seq data [55] over all *C. albicans* transcripts, a coverage matrix was computed using the deepTools2 “computeMatrix” function ([52]; reference-point -a 1000 -b 1000 -bs 5). Transcripts from *C. albicans* were extracted by the “transcripts” function of the GenomicRanges package [54] using the current genomic annotation from the *C. albicans* assembly 22 (version A22-s07-m01-r88; http://www.candidagenome.org/). MNase-seq reads were extended to 100 bp prior to coverage plotting as described in the original publication [55].

Differential read coverage (log2-ratio) in nucleosome-free ATAC-seq peaks between H_2_O_2_-treated and non-treated samples was analyzed by the deepTools2 “bamCompare” function ([52]; settings: --normalizeUsing CPM -bs 5 –e --scaleFactorsMethod None). For downstream analysis, this output was first used to compute a coverage matrix with the deepTools2 “computeMatrix” tool [52] as described above across promoter regions (−1000/+200 bp with respect to the TSS) of all *C. albicans* transcripts. Heat map and coverage plots were generated using the “plotHeatmap” function from deepTools2 [52] using *K*-means clustering (settings: -kmeans 4). The full analysis data and the RNA-seq regulation of clustered genes are provided in Appendix A. Coverage regions from cluster 1 and cluster 4 were further subjected to gene ontology (GO) term analysis using the “enrichGO” function from the clusterProfiler package ([56]; settings: ont = “BP”, qvalueCutoff = 0.05, readable = TRUE). The GO term analysis result was plotted using the clusterProfiler “dotplot” function after merging redundant GO terms using the “simplify” function from the same package (settings: cutoff = 0.7) [56].

#### 2.3.3. Peak Calling and Genomic Annotation of ATAC-seq Peaks

Peak-calling for each individual sample was done with MACS2 v2.1.2 using “callpeak” ([57]; settings: -f BAMPE -g 14521502). The aligned read fragments from the gDNA ATAC-seq libraries were merged into one BAM file using the SAMtools “merge” function [58] and used as a background control for peak calling. Peaks from all samples and replicates were merged and converted to the GFF file format for read counting using htseq-count ([59]; settings: -f bam-s no-t peak) and differential ATAC-seq peak analysis (see below). The reproducibility of called ATAC-seq peaks among biological replicates was analyzed via principal component analysis (PCA) using the “prcomp” function from the R stats package (https://www.rdocumentation.org/packages/stats). For the ATAC-seq peak annotation, the genomic annotation from *C. albicans* assembly 22 (version A22-s07-m01-r88) was downloaded from the *Candia* Genome Database (CGD; http://www.candidagenome.org/) and was merged with the 5′UTR (untranslated region) and TSS annotations of *C. albicans*, which were retrieved from the yeast transcription start site database (YeasTSS) (http://www.yeastss.org/). This file was further used to create a TxDb object using the GenomicFeatures “makeTxDbFromGFF” function ([54]; settings: dataSource = “CGD”, organism = “Candida albicans SC5314”, circ_seqs = “Ca22chrM_C_albicans_SC5314”). Peak annotation was done using the ChIPseeker Bioconductor package [60] with the “annotatePeak” function (options: tssRegion = c(−2000, 0), genomicAnnotationPriority = c(“Promoter”, “5UTR”, “3UTR”, “Exon”, “Intron”, “Downstream”, “Intergenic”)). The distribution and overlaps of peak annotations among genomic features were visualized with the “upsetplot” function from the ChIPseeker package. The ATAC-seq peak location across all chromosomes and the fold change in peak signals between H_2_O_2_-treated and non-treated samples were visualized using the karyoploteR package [61].

#### 2.3.4. Analysis of Differential ATAC-seq Peaks

Differential accessible peak detection was done using the edgeR package [62] with the generalized linear model (GLM) approach. Since replicate three clustered away from the other two replicates in the PCA analysis (Appendix A), the GLM approach was combined with batch effect correction. Appendix A contains a complete list of detected ATAC-seq peaks and the edgeR analysis result. Appendix A presents the ATAC-seq results overlaid by the RNA-seq datasets. GO term analysis of genes with significantly increased peak signal (log2-fold change > 0, false discovery rate (FDR) < 0.05) in upstream regions upon H_2_O_2_ stress was performed with the “enrichGO” function from the clusterProfiler package [56] as described above. 

#### 2.3.5. Motif Search

Nucleosome-free ATAC-seq peaks with increased signal in the H_2_O_2_-treated samples (log2-fold change > 0, FDR < 0.05) from the edgeR analysis were further subjected to motif search analysis using the MEME suite (v5.1.0; [63]; http://meme-suite.org/). First, the previously identified *C. albicans* Cap1 regulon [45] was used to generate the position weight matrix for Cap1 with the MEME web application (settings: 1 occurrence per sequence, motif width = 8). This was further used as input for the MEME FIMO tool [64] to analyze Cap1 binding sites in the upregulated nucleosome-free ATAC-seq peaks in H_2_O_2_-treated samples (settings: –parse-genomic-coord). The enrichment of the identified Cap1 binding sites was re-formatted to the BED file format and further used to compute a coverage matrix with the “computeMatrix” function from deepTools2 ([52]; settings: reference-point, -b 500 -a 500 -bs 5 --referencePoint center) in nucleosome-free ATAC-seq peaks from H_2_O_2_-treated and non-treated samples. For de novo motif analysis, upregulated nucleosome-free ATAC-seq peaks during oxidative stress were filtered based on their occurrence within the 1 kb upstream region of a gene and 500 bp of maximum peak width. The resulting 184 regulated peaks were used for de novo motif search using the MEME suite online tool DREME ([65]; parameters: E-value threshold 1; search given strand only). 

#### 2.3.6. Data Plotting

Unless otherwise stated, plots were done with the “ggplot2” package in R [66].

### 2.4. External Datasets

Our previously published [16] RNA-seq data from *C. albicans* in response to 30 min H_2_O_2_ treatment (Gene Expression Omnibus (GEO) accession number GSE73409) were used to compare ATAC-seq peaks with increased signal in H_2_O_2_-treated samples with genes upregulated in response to oxidative stress. MNase-seq data ([55]; described above) was downloaded from the SRA (SRR059732).

### 2.5. Code Availability

The entire bioinformatics analysis pipeline is freely accessible on github: https://github.com/tschemic/ATACseq_analysis.

### 2.6. Data Availability

ATAC-seq data have been deposited at the GEO under the accession number GSE156582. 

## 3. Results and Discussion

### 3.1. ATAC-Seq in C. albicans Reflects Nucleosomal Organization Genome-Wide

We adapted and optimized an ATAC-seq method originally developed for the fungi *S. cerevisiae* and *Schizosaccharomyces pombe* [31] to study alterations in chromatin landscapes upon oxidative stress in the human fungal pathogen *C. albicans*. Therefore, logarithmically growing *C. albicans* cells were either treated with 1.6 mM H_2_O_2_ for 15 min or left untreated (see Materials and Methods for details). After spheroplasting, cells were then subjected to sample preparation for ATAC-seq analysis. To correct for the known sequence bias of the Tn5 transposase [47], naked genomic DNA (gDNA) prepared from YPD-grown *C. albicans* was included in the workflow (Figure 1A; see Materials and Methods for details). Three biological replicates were prepared for each condition. Analysis of PCR-amplified ATAC-seq libraries from intact fungal cell nuclei (chromatin) showed a distinct fragment size distribution reminiscent of nucleosomal periodicity. Importantly, ATAC-seq libraries prepared from naked gDNA had a less complex fragment distribution as they lacked nucleosomal patterns (Figure 1B). The same trend was observed after 75 bp paired-end sequencing (PES), where the fragment length distribution of PES reads from fungal chromatin showed a predominant peak of shorter fragments around <150 bp, representing putative nucleosome-free regions. Another peak with an average fragment length of 200 bp reflected DNA regions occupied by one nucleosome (Figure 1C; [49]). Additionally, a weak enrichment for di-nucleosomal peaks was detected. Again, ATAC-seq libraries from gDNA did not reveal nucleosomal presence (Figure 1C). Instead, the majority of PES read fragments were below 150 bp, which was consistent with the fragment length distribution of Tn5-generated DNA sequencing libraries [67]. Notably, the number of mapped reads was evenly distributed among the *C. albicans* chromosomes, thus reflecting total chromosome sizes and suggesting no chromosomal bias (Appendix A). Inspection of ATAC-seq reads aligned to the *C. albicans* genome further revealed a distinct read coverage profile of accessible chromatin regions in nuclear chromatin purified from H_2_O_2_-treated (H_2_O_2_) and non-treated (YPD) cells, which was not apparent in gDNA libraries (Figure 1D shows a region of chromosome 1 as an example).

To further test the robustness of our ATAC-seq workflow, we first selected ATAC-seq PES read fragments below 100 bp, which correspond to putative nucleosome-free genomic regions [30]. These data were subjected to nucleosomal occupancy prediction using the NucleoATAC tool [31]. First, read coverage profiles of nucleosome-free ATAC-seq peaks and NucleoATAC-called nucleosomes were inspected at an example locus on chromosome 6 containing four open reading frames (ORFs) (Figure 2A). Pronounced nucleosome-free ATAC-seq peaks, flanked by nucleosomal signals as predicted by NucleoATAC, were detected in upstream promoter regions (Figure 2A, black arrows).

Moreover, nucleosomal positions predicted by NucleoATAC correlated very well with published *C. albicans* MNase-seq data [55] (Figure 2B). By averaging the read coverage of nucleosome-free and mononucleosmal ATAC-seq signals across all *C. albicans* transcripts relative to the transcription start site (TSS), we further observed an enrichment of nucleosome-free ATAC-seq reads adjacent to the TSS. Mononucleosomal ATAC-seq signals were enriched up- and downstream of the nucleosome-free peak, reflecting a well-positioned +1 nucleosome and a typical nucleosomal organization of canonical active gene promoters [26,68] (Figure 2B). Taken together, ATAC-seq libraries generated from *C. albicans* native chromatin share typical features of mammalian and other fungal (*S. cerevisiae* and *Schizosaccharomyces pombe*) ATAC-seq samples [30,31]. Hence, the approach is suitable to probe chromatin accessibility and nucleosomal positioning in *C. albicans* and their changes during stress response.

### 3.2. ATAC-Seq Detects Genome-Wide Changes in Chromatin Accessibility after Oxidative Stress

Next, we aimed to assess whether ATAC-seq in *C. albicans* is able to measure changes in chromatin accessibility at genomic regions associated with the oxidative stress response. Therefore, we analyzed the differential read coverage of nucleosome-free ATAC-seq peaks in H_2_O_2_-treated and non-treated cells. Then, we further clustered all *C. albicans* promoter regions (−1000/+200 bp upstream of the TSS) based on their differential read coverage profile using *K*-means clustering (Figure 2C). Genomic regions with a moderate increase in nucleosome-free ATAC-seq peak signals around the TSS were enriched in cluster 2, while cluster 3 showed a subtle increase in ATAC-seq read coverage more distal relative to the TSS (Figure 2C,D, orange and light blue lines, respectively, in panel D). In response to oxidative stress, the most striking increase in chromatin accessibility (i.e., increased nucleosome-free ATAC-seq reads) proximal to the TSS and at least 1 kb upstream of the TSS, was observed for loci contained in cluster 1 (Figure 2C,D, red line in panel D). GO term enrichment analysis of promoter-associated genes from cluster 1 showed that around 12% of the genes in cluster 1 are related to the oxidative stress response (Figure 2E). Cluster 4 included regions with decreased chromatin accessibility upstream of the TSS in response to H_2_O_2_ treatment (Figure 2C,D, dark blue line in panel D). Genes from this cluster were involved in ribosome- and translation-related processes (Appendix A), most of which are often downregulated during fungal stress adaptation [69,70,71,72,73].

The degree of chromatin accessibility is typical for genomic regions experiencing distinct transcriptional activities. For instance, the TSS of active gene promoters and enhancers is associated with increased chromatin accessibility relative to inactive elements and heterochromatic regions [28,74,75,76]. Hence, the combination of ATAC-seq and RNA-seq can provide highly useful insights into the chronological dynamics of changing chromatin accessibility and subsequent transcriptional responses [42,77]. In a previous study, we performed RNA-seq analysis in *C. albicans* in response to H_2_O_2_ treatment [16]. For RNA-seq, cells were treated with 1.6 mM H_2_O_2_ for 30 min, while cells for ATAC-seq were exposed to 1.6 mM H_2_O_2_ only for a total of 20 min. Therefore, we used the RNA-seq dataset to assess the chronological dynamics of chromatin accessibility and transcription.

When assessing the log2-fold change from our RNA-seq dataset of genes with increased chromatin accessibility upon H_2_O_2_ treatment in upstream genomic regions (cluster 1), we observed that transcription of the majority of cluster 1 genes was also upregulated in response to H_2_O_2_—321 genes mapped to cluster 1, of which 309 were also present in the RNA-seq dataset. Out of those 309 genes, 200 were transcriptionally upregulated upon H_2_O_2_ treatment (Appendix A). For instance, *CAT1*, which encodes the H_2_O_2_-detoxifying catalase [78], as well as the oxidative stress-related Cap1 transcription factor and its known targets *OYE*32 and *CIP1* [79], were among genes showing remarkably increased chromatin accessibility and transcriptional induction in H_2_O_2_-treated cells (Figure 2F). In addition, *ICL1* and *MSL1* encoding for glyoxylate cycle enzymes [80] showed increased ATAC-seq peak signals, reflecting their robust transcriptional induction upon oxidative stress (Figure 2F). Similarly, genes with decreased ATAC-seq read coverage upstream of their TSS were likewise repressed upon stress (Appendix A). Hence, changes in ATAC-seq read coverage are predictive of alterations in gene expression.

To further dissect significant changes in chromatin accessibility, nucleosome-free ATAC-seq signals were subjected to peak calling with ATAC-seq libraries from gDNA as background noise control (see Material and Methods for details). The genomic locations of the majority of called nucleosome-free ATAC-seq peaks were located within gene promoters, exons and 5′ untranslated regions (UTRs) (Figure 3A) reflecting a usual distribution of genomic features among called ATAC-seq peaks [81].

Of note, substantial peak overlaps between these three genomic regions were observed, indicating peaks stretching from promoter regions into the 5′ UTRs and the first exon. Interestingly, a previous study observed ATAC-seq signals accumulating towards sub-telomeric regions [39]. Thus, we further inspected the genome-wide distribution of differentially enriched peaks, but we did not detect a bias toward specific chromosomal regions including telomeres (Figure 3B).

Nucleosome-free ATAC-seq peaks in H_2_O_2_-treated and non-treated cells from all three biological replicates were then subjected to principal component analysis (PCA) based on the presence and absence of called peaks. Samples clustered according to their growth conditions, suggesting that the peak calling of the ATAC-seq workflow was able to identify alterations in chromatin accessibility in response to oxidative stress (Appendix A). Notably, the PCA also revealed variations among biological replicates, as replicate three from stressed and non-stressed cells was separated from the other two replicates by the first principal component (Appendix A, PC1). Hence, we also applied a batch effect correction during differential peak analysis of called ATAC-seq peaks (see Material and Methods for details). In total, 1092 nucleosome-free ATAC-seq peaks with significant signal alterations (FDR < 0.05) in response to oxidative stress were detected (Appendix A). Peaks were annotated to the next closest downstream gene and the resulting dataset was overlaid with the RNA-seq data to correlate differentially regulated ATAC-seq peaks with transcriptional changes. In total, 468 ATAC-seq peaks were upregulated in H_2_O_2_-treated cells and located within the 2 kb upstream region of the annotated gene, with 298 peaks detected upstream of genes with increased transcription after oxidative stress (Figure 3C, Appendix A). These data demonstrate that more than 60% of genomic regions showing increased nucleosome-free ATAC-seq peak signals within a 2 kb upstream range were associated with transcriptional induction. For instance, as already observed by comparing ATAC-seq read coverages between the treatment groups, oxidative stress altered chromatin accessibility of promoter regions and transcript induction of genes including *CAT1*, *ICL1*, *OYE32*, and *TSA1*, the latter encoding an antioxidative protein [82] (Figure 3C). Accordingly, genes downstream of upregulated ATAC-seq peaks represented biological processes involving oxidative stress responses. Interestingly, gene promoters of almost 90 genes, including some encoding uncharacterized proteins, displayed significantly elevated chromatin accessibility in promoter regions during H_2_O_2_ treatment without alterations in gene expression. For example, gene *C2_09880C*, encoding for a putative DNA-binding factor, was among the top 20 genes with increased chromatin accessibility upon oxidative stress. However, no significant alteration in transcription was detected in the corresponding RNA-seq dataset (Figure 3C and Appendix A). Such cases could represent putative regulators that are primed for transcriptional control in response to additional signaling inputs [83]. Generally, out of a total of 5284 ATAC-seq peaks detected within the 2 kb upstream regions of annotated genes, 3174 peaks showed no ATAC-seq signal alterations in response to oxidative stress, despite significant transcriptional regulation of the downstream gene after H_2_O_2_ treatment (Appendix A). These discrepancies might reflect distinct regulatory dynamics between the establishment of permissive open chromatin and highly transient transcriptional responses often seen in stress adaptation [72]. Indeed, the samples for RNA-seq data were collected after 30 min of oxidative stress, while the ATAC-seq dataset was derived from cells exposed to H_2_O_2_ for a total of 20 min. In summary, these data demonstrate that ATAC-seq detects changes in chromatin accessibility linked to gene expression control during fungal stress adaptation. Moreover, it reflects highly dynamic transcriptional responses and could be used to predict novel regulators that are primed for further transcriptional alterations during environmental adaptation.

### 3.3. Oxidative Stress-Responsive ATAC-Seq Peaks Are Enriched for Cap1 Binding Sites

Despite assessing the chromatin architecture at different genomic features, ATAC-seq has been used to detect sequence motifs used by transcriptional regulators in ATAC-seq peak regions [32,39]. This is of special interest for the identification of as-yet-unknown transcriptional regulators or even regulatory networks governing transcriptional responses. Indeed, regulatory sequences were identified in the malaria parasite *Plasmodium falciparum* during intra-erythrocytic development [39]. Hence, ATAC-seq can yield new insights into transcriptional networks and their dynamic behavior during environmental adaptations. For example, major virulence traits of *C. albicans*, such as morphological transitions, are under the control of multi-layer genetic regulatory networks engaging both transcription factors and chromatin modifiers [20,84,85]. Hence, we tested whether our ATAC-seq data were suitable for motif discovery in *C. albicans*. As mentioned briefly above, the Cap1 transcription factor is the key driver of transcriptional induction during the oxidative stress response [79]. Its *cis*-acting sequence motif emerged from an earlier study [45] (Figure 4A).

When we analyzed the enrichment of published Cap1-binding sites in nucleosome-free ATAC-seq peaks in H_2_O_2_-treated cells, we observed that ATAC-seq read signals were indeed highly increased around the Cap1 motif (Figure 4B). Importantly, a de novo motif search identified Cap1 binding sites appearing in ATAC-seq peaks that showed increased intensity after oxidative stress (Figure 4C, motif 3). Of note, some other motifs contained repetitive deoxyadenosine (poly-dA) stretches (Figure 4C, motif 1). While these might be less relevant for the recruitment of transcription factors, such signals can be explained by the preferred occurrence of poly(dA:dT) tracts in nucleosome-depleted regulatory regions, such as TATA elements upstream of the TSS, and the tendency of A-rich elements in TATA-less promoters in *S. cerevisiae* [86,87]. In addition, GGTTT/AAACC motifs are overrepresented in ATAC-seq data due to a binding bias of the Tn5 transposase to such regions [88]. These data suggest that a more stringent correction of the Tn5 sequence bias might be beneficial for a de novo motif discovery with higher precision. Notably, tools for advanced sequence bias correction of enzymatically prepared sequencing libraries, including DNase-seq, MNase-seq, and ATAC-seq, are available [88,89].

In conclusion, we here present an integrative workflow for ATAC-seq in the human fungal pathogen *C. albicans.* ATAC-seq offers a new robust tool to capture temporal changes in chromatin landscapes that are tightly connected to transcript abundance during fungal adaptation to stress. In addition, novel regulators that are not subject to transcriptional control at the time of sample collection might be identified based on chromatin accessibility signatures. This might aid in capturing a broader picture of highly dynamic cellular adaptations where sample drawing is limited. Finally, the combination of ATAC-seq with additional datasets has the potential to predict yet-unknown regulatory sequences and transcription factor binding sites that dictate transcriptional reprogramming and, thus, may be crucial to fungal survival upon stress encounter. Finally, the present workflow might be applicable to in vivo ATAC-seq approaches, since we were already able to obtain meaningful ATAC-seq data from as little as 50,000 *C. albicans* spheroplasts. The re-isolation of comparable fungal cell numbers from host model systems, such as mice, is feasible and thus, ATAC-seq might facilitate studies that elucidate fungal responses and adaptation to host defense in a tissue-specific manner. Such data will not only help to better understand fungal tissue tropism but also offer to identify new fungal genes amenable to therapeutic intervention.

## Figures and Tables

**Figure 1 jof-06-00182-f001:**
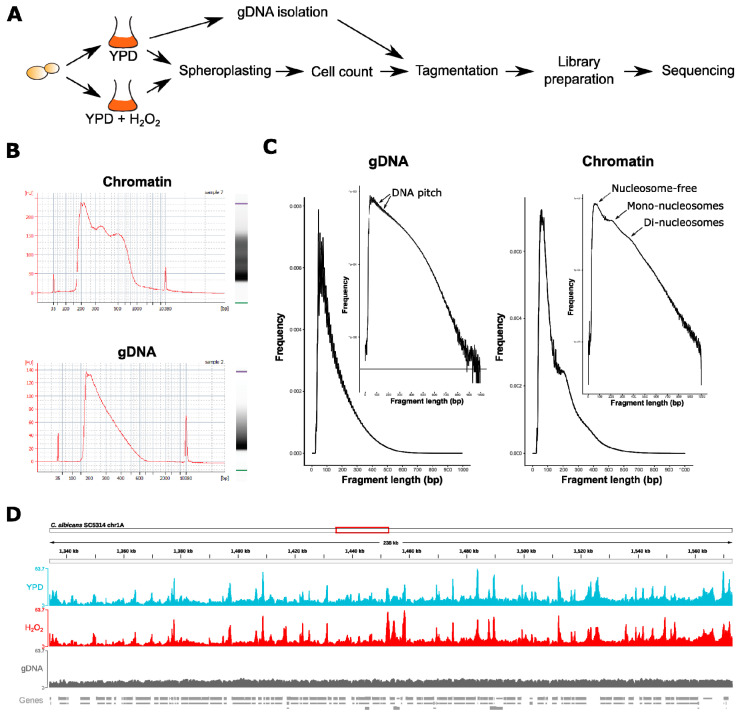
Quality control of assay for transposase-accessible chromatin using sequencing (ATAC-seq) libraries from *Candida albicans* treated or non-treated with H_2_O_2_. (**A**) Experimental set-up for ATAC-seq profiling of 5 × 10^6^
*C. albicans* (see Materials and Methods for details). (**B**) Bioanalyzer electrophoretic profiles and gel image from ATAC-seq libraries prepared from fungal chromatin (isolated nuclei) or naked gDNA. The x-axis from the electropherogram depicts the fragment size distribution (bp) of tagmented samples and the y-axis represents the abundance. A distinct nucleosomal pattern (mono-, di-, and tri-nucleosomes) is visible in chromatin samples, but not in tagmented gDNA samples. (**C**) Fragment length distribution of ATAC-seq reads from gDNA and nuclei prepared from YPD (1% yeast extract, 2% peptone, 2% glucose)-grown cells. The fragment length in bp (x-axis) from one representative biological replicate from each group is plotted against its frequency (y-axis). The graph insert shows the log-transformed histogram. Signals originating from the DNA helical pitch [31] are emphasized by arrows on the gDNA plot. Arrows on the chromatin plot indicate nucleosome-free ATAC-seq read fragments and nucleosome occupied read fragments. (**D**) Integrative Genomics Viewer (IGV) browser snap-shot from a 238 kb region of the *C. albicans* chromosome 1A of ATAC-seq reads from aligned BAM files. The biological replicates from each sample group were pooled. Genes are depicted as gray boxes below the sequencing read coverage tracks.

**Figure 2 jof-06-00182-f002:**
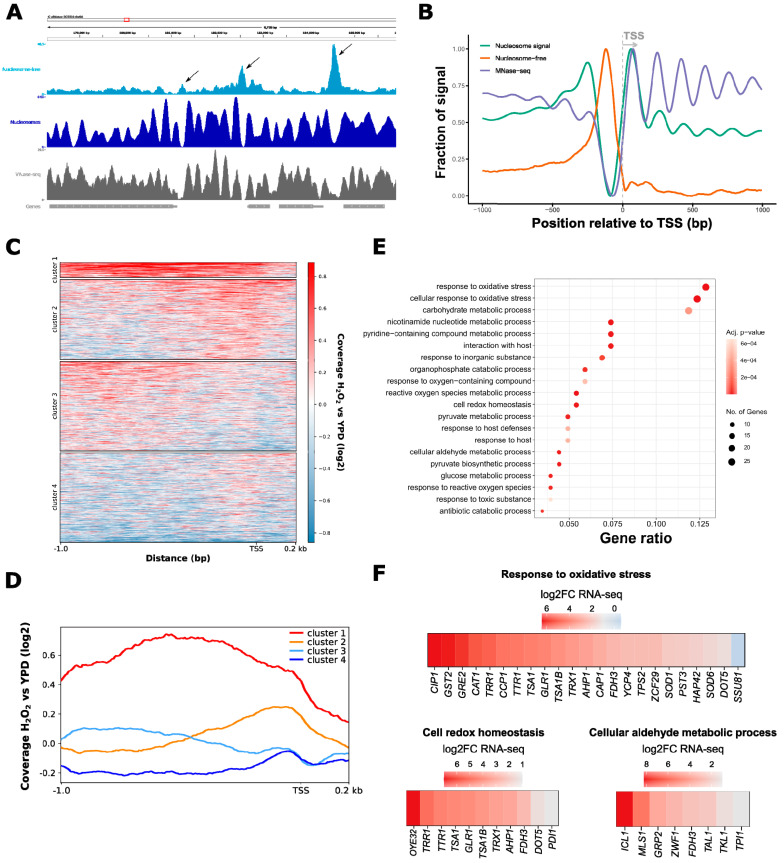
Chromatin accessibility is increased at promoters associated with oxidative stress genes. (**A**,**B**) ATAC-seq analysis in *C. albicans* allows for nucleosomal occupancy prediction. Comparison of coverage profiles of nucleosome-free ATAC-seq read signals with mono-nucleosomal position prediction and MNase-seq data [55] at a region on chromosome 6 (**A**) or across all *C. albicans* transcriptional start sites (+/−1000 bp relative to the transcription start site (TSS)) (**B**). Data represent ATAC-seq samples from YPD grown cells and pooled biological replicates (see Materials and Methods for details). In panel A, genes are depicted as gray boxes with white arrows indicating the direction of transcription. The black arrows in the top IGV track indicate nucleosome-free ATAC-seq signals flanked by nucleosomes. (**C**,**D**) *K*-means clustering of differential ATAC-seq nucleosome-free read coverage (log2-fold change) of H_2_O_2_-treated (H_2_O_2_) vs. non-treated (YPD) cells represented as heat maps (**C**) or read coverage profiles across all *C. albicans* promoters (−1000 bp/+200 bp relative to the TSS). (**E**,**F**) Gene ontology (GO) term analysis of cluster 1 from panel C and D represented as a dot plot (**E**) and heat maps (**F**). Genomic regions represented in cluster 1 were annotated to the next downstream gene. The gene ratio indicates the proportion of genes enriched in this GO term relative to input genes. The color scale indicates the adjusted *p*-value of the GO enrichment analysis. Heat maps show genes from the indicated GO term category with the color scale representing the log2-fold change in transcription (from RNA-seq data) in response to oxidative stress.

**Figure 3 jof-06-00182-f003:**
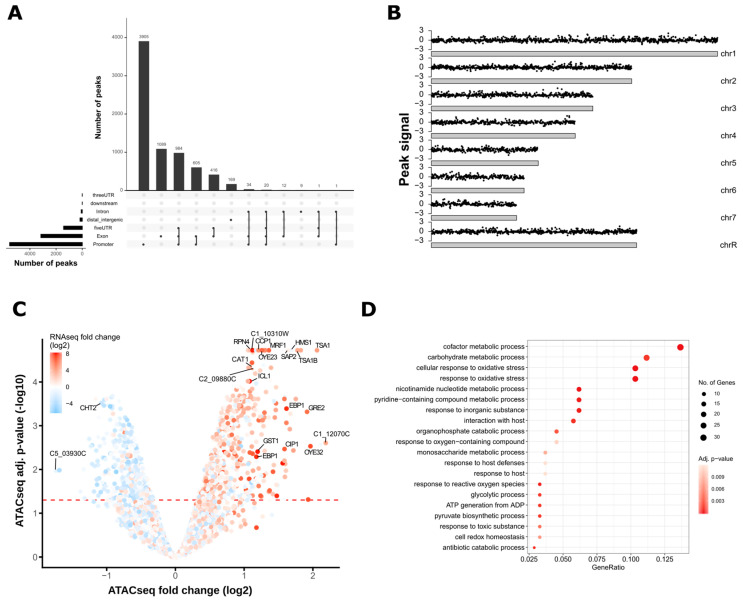
ATAC-seq nucleosome-free peak annotation and differential ATAC-seq peak analysis. (**A**) The majority of called ATAC-seq nucleosome-free peaks are within gene promoters. The vertical bars of the UpSet plot represent the number of peaks spanning one or more specific genomic regions (overlaps of genomic regions are indicated by connection lines between the black dots). For instance, the first bar indicates the number of peaks occurring exclusively in promoters, while the third bar represents peaks spanning promoter, 5′ UTR (untranslated regions) and exonic regions. The horizontal bars show the total number of peaks for each genomic annotation, independent of overlaps. (**B**) Karyoplots showing the genomic peak location on each chromosome of differential ATAC-seq nucleosome-free read signals (log2-fold change) of H_2_O_2_-treated vs. non-treated cells. The log2-fold change is depicted on the y-axis and the x-axis represents each chromosome of the *C. albicans* genome. (**C**) Increased ATAC-seq nucleosome-free peak signals correlate with increased transcript levels. Volcano plot depicting the differential peak signal analysis of nucleosome-free ATAC-seq peak signals between H_2_O_2_-treated and non-treated cells. The y-axis represents the negative log10 adjusted *p*-value (false discovery rate, FDR) and the x-axis depicts the log2 fold change between H_2_O_2_ treatment vs. no treatment. The horizontal dashed red line indicates an FDR of 0.05. Each dot represents one peak annotated to the next downstream gene, which is overlaid with a color gradient indicating the log2-fold change (H_2_O_2_ treatment vs. no treatment) from RNA-seq data. (**D**) Peaks with increased ATAC-seq nucleosome-free signals in H_2_O_2_-treated cells are upstream of genes related to the oxidative stress response. The gene ratio indicates the proportion of genes enriched in the indicated GO term relative to input genes with increased chromatin accessibility after H_2_O_2_ stress. The color scale represents the adjusted *p*-values of the GO enrichment analysis.

**Figure 4 jof-06-00182-f004:**
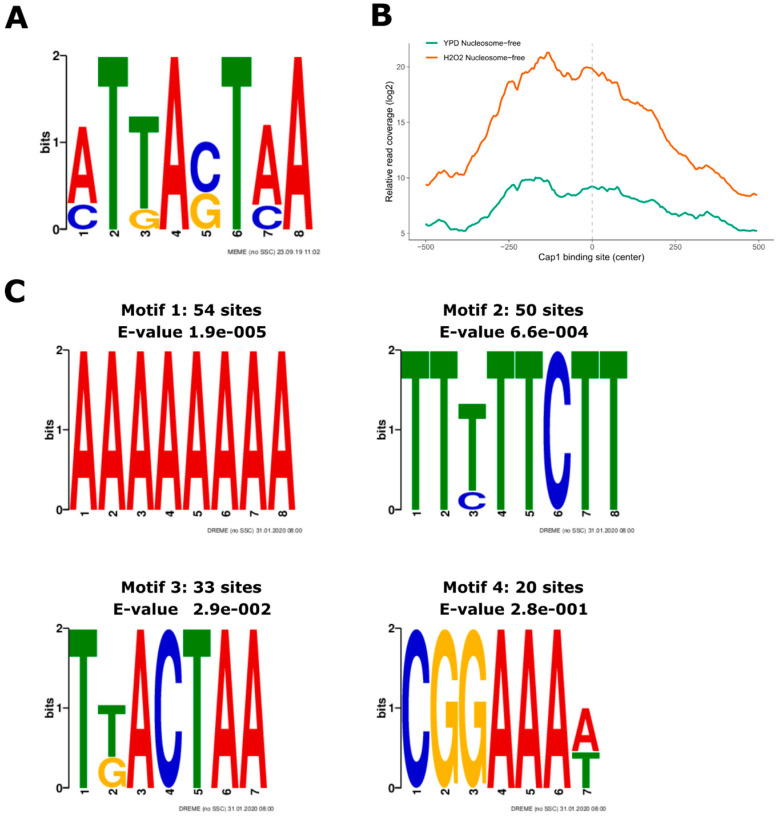
De novo motif search in oxidative stress-responsive nucleosome-free ATAC-seq peaks. (**A**) Previously identified Cap1 motif [45] used for enrichment analysis in nucleosome-free ATAC-seq peaks with increased signal in H_2_O_2_-treated cells. (**B**) Cap1 binding sites are enriched for nucleosome-free ATAC-seq fragments in response to H_2_O_2_. Published Cap1 binding sites [45] were used for enrichment analysis within nucleosome-free ATAC-seq peaks. Read coverage in H_2_O_2_-treated and untreated cells around the Cap1 binding sites is shown. (**C**) Logos of position weight matrices from de novo identified motifs occurring in nucleosome-free ATAC-seq peaks in H_2_O_2_-treated cells (see Material and Methods for details). The number of positive matches among the input sequences and the e-value from the MEME suite DREME tool are indicated above each logo. The top four logos are presented.

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
