# Peer review of "ATAC-Seq Identifies Chromatin Landscapes Linked to the Regulation of Oxidative Stress in the Human Fungal Pathogen Candida albicans"

_jof, 2020, doi:10.3390/jof6030182_

Round 1

Reviewer 1 Report

In this study, Jenull et al explore C. albicans chromatin accessibility following H202 treatment, a condition mimicking the reactive oxygen species by the host’s immune cells. To this end, the authors adapted to C. albicans the ATAC-seq protocol, a technique widely used to analyse chromatin accessibility genome-wide. 

Using this genome-wide approach, the authors demonstrate that :

  1. ATAC-seq is a reliable technique to measure chromatin accessibility in C. albicans
  2. H2O2 treatment leads to increased chromatin accessibility in genes associated with the oxidative response. 
  3. Genomic regions with pronounced ATAC-seq signal are enriched in binding sites for Cap1, a transcriptional regulator involved in the oxidative stress response. 

The article is clearly written; the experiments are overall carefully performed and nicely presented. The data are of high quality. In summary, this study will serve as a useful resource to the C. albicans community and it will help our understanding of how chromatin structure regulates C. albicans adaptation. 

I have one question:

The authors compare differences in chromatin accessibility with gene expression difference by using the ATAC-seq data generated in this study and the RNA-seq data generated in a previous study 

The author state that this comparison is possible because “the H2O2 treatment (between the two different studies) was identical” (Line 434).

However, in line 515 the authors state that the conditions of the two experiments are different “Notably, the here used RNA-seq data was collected after 30 minutes of oxidative stress, while the ATAC-seq data set was derived from cells exposed to hydrogen peroxide for 20 minutes” The authors should explain this apparent discrepancy.

Minor comments:

  1. The introduction would benefit from a brief description of the ATAC-seq technique and an explanation of the underlying principle.
  2. Figure 2E and F are partly redundant.
  3. Figure 2: Legend too long. 
  4. Line 21: C. albcians- The spelling should be corrected to C. albicans
  5. Throughout the text: C. albicans should be changed to italics
  6. Line 40: [4],[5] correct Reference format
  7. Line 515: the here used RNA-seq data- This sentence should be corrected

Author Response

In this study, Jenull et al explore C. albicans chromatin accessibility following H202 treatment, a condition mimicking the reactive oxygen species by the host’s immune cells. To this end, the authors adapted to C. albicans the ATAC-seq protocol, a technique widely used to analyse chromatin accessibility genome-wide. 

Using this genome-wide approach, the authors demonstrate that :

  1. ATAC-seq is a reliable technique to measure chromatin accessibility in C. albicans
  2. H2O2 treatment leads to increased chromatin accessibility in genes associated with the oxidative response. 
  3. Genomic regions with pronounced ATAC-seq signal are enriched in binding sites for Cap1, a transcriptional regulator involved in the oxidative stress response. 

The article is clearly written; the experiments are overall carefully performed and nicely presented. The data are of high quality. In summary, this study will serve as a useful resource to the C. albicans community and it will help our understanding of how chromatin structure regulates C. albicans adaptation. 

Response 1: We appreciate this positive view and thank Reviewer 1 for her/his insightful feedback, which helped us to improve the manuscript. Please find our answers to each comment point by point below.

I have one question:

The authors compare differences in chromatin accessibility with gene expression difference by using the ATAC-seq data generated in this study and the RNA-seq data generated in a previous study 

The author state that this comparison is possible because “the H2O2 treatment (between the two different studies) was identical” (Line 434).

However, in line 515 the authors state that the conditions of the two experiments are different “Notably, the here used RNA-seq data was collected after 30 minutes of oxidative stress, while the ATAC-seq data set was derived from cells exposed to hydrogen peroxide for 20 minutes” The authors should explain this apparent discrepancy.

Response 2: Yes indeed and we apologize for the confusion. The data sets were derived from conditions as stated in line 530 in the revised manuscript (ATAC-seq with a total H2O2 exposure of 20 minutes and RNA-seq with a total H2O2 exposure of 30 minutes). We corrected the statement from line 434 accordingly, which reads now (line 433-436 in the revised manuscript):

“For RNA-seq, cells were treated with 1.6 mM H2O2 for 30 minutes, while cells for ATAC-seq were exposed to 1.6 mM H2O2 only for a total of 20 minutes. Therefore, we used the RNA-seq dataset to assess the chronological dynamics of chromatin accessibility and transcription.”

Minor comments:

  1. The introduction would benefit from a brief description of the ATAC-seq technique and an explanation of the underlying principle.

Response 3: We apologize for not giving enough background information about the method. We have thus added the following sentence into the introduction (line 75-80 in the revised manuscript):

“ATAC-seq employs a hyperactive Tn5 transposase loaded with sequencing adapters, which are inserted into accessible chromatin sites, causing fragmentation and tagging of chromatin DNA referred to as tagmentation. This happens preferably at genomic regions with open, accessible chromatin, since transposition events into condensed chromatin are less likely.”

  1. Figure 2E and F are partly redundant.
  2. Figure 2: Legend too long. 

Response 4: Thank you for raising this valid point. We agree that the legend contained too much information and we therefore shortened it. The new legend now reads as follows (line 376-391 in the revised manuscript):

Figure 2. Chromatin accessibility is increased at promoters associated with oxidative stress genes. A-B) ATAC-seq analysis in C. albicans allows for nucleosomal occupancy prediction. Comparison of coverage profiles of nucleosome-free ATAC-seq read signals with mono-nucleosomal position prediction and MNase-seq data [55] at a region on chromosome 6 (A) or across all C. albicans transcriptional start sites (+/- 1000 bp relative to TSS) (B). Data represent ATAC-seq samples from YPD grown cells and pooled biological replicates (see Materials & Methods for details). In panel A, genes are depicted as grey boxes below with white arrows indicating the direction of transcription. The black arrows in the top IGV track indicate nucleosome-free ATAC-seq signals flanked by nucleosomes. C-D) K-means clustering of differential ATAC-seq nucleosome-free read coverage (log2-fold change) of H2O2-treated (H2O2) vs non-treated (YPD) cells represented as heatmaps (C) or read coverage profiles across all C. albicans promoters (-1000bp/+200bp relative to the TSS). E-F) GO term analysis of cluster 1 from panel C-D represented as dotplot (E) and heatmaps (F). Genomic regions represented in cluster 1 were annotated to the next downstream gene. The Gene Ratio indicates the proportion of genes enriched in this GO term relative to input genes. The color scale indicates the adjusted p-value of the GO enrichment analysis. Heatmaps show genes from the indicated GO term category with the color scale representing the log2-fold change in transcription (from RNA-seq data) in response to oxidative stress.”

  1. Line 21: C. albcians- The spelling should be corrected to C. albicans
  2. Throughout the text: C. albicans should be changed to italics
  3. Line 40: [4],[5] correct Reference format

Response 5: We apologize for these typos – all corrected.

  1. Line 515: the here used RNA-seq data- This sentence should be corrected

Response 6: As outlined above, we clarified this misunderstanding.

Reviewer 2 Report

Overall, this is a well written manuscript. The introduction clearly justifies the approach and the data demonstrate that the protocol works. The experimental work and the bioinformatic analyses seem to be done well.

I have only minor comments:

Figure 2F is rather hard to interpret, does not really add anything to the manuscript and it is not obviously supportive of the text after which it is cited: “, we observed that transcription of the majority of cluster 1 genes was also upregulated in response to H2O2 (Figure 2F).”. I think your point can be more clearly made by a heatmap displaying the RNA-seq based gene expression data for all of the genes in cluster 1 or by simply making a supplementary table to present the data.

Along these same lines, on lines 439 (referring to Figure 2F) and 499 (referring to figure 3C) you state that the “majority of genes” show concordant results between the ATAC-seq and RNA-seq (open chromatin and increased gene expression) following the H2O2 treatment. This is not precise enough for a scientific publication. Please specifically state the exact numbers. Perhaps this information can be included in a supplemental table.

It would also be very useful, in the interest of full disclosure, to include another Supplementary table with the RNA-seq data for all of the genes for which you have identified an ATAC-seq peak upstream. In this new table, you should clearly mark whether each the peak/gene is concordant or discordant with the RNA-seq data and report these numbers in the text. Since this is the first report of ATAC-seq in Candida albicans, this information would be useful as a comparator for future ATAC-seq studies that are sure to come.

Line 434: treatement was NOT identical and you acknowledge this in line 517. Please change to make it consistent.
